# Effect of Self-Management Support for Elderly People Post-Stroke: A Systematic Review

**DOI:** 10.3390/geriatrics5020038

**Published:** 2020-06-18

**Authors:** Sedsel Kristine Stage Pedersen, Susanne Lillelund Sørensen, Henriette Holm Stabel, Iris Brunner, Hanne Pallesen

**Affiliations:** 1Hammel Neurorehabilitation Centre and University Research Clinic, University of Aarhus, 8450 Hammel, Denmark; Susanne.Lillelund@rm.dk (S.L.S.); Henriette.Holm.Stabel@midt.rm.dk (H.H.S.); Iris.Brunner@rm.dk (I.B.); hannpall@rm.dk (H.P.); 2Department of Clinical Medicine, Faculty of Health, Aarhus University, 8000 Aarhus, Denmark

**Keywords:** self-management, stroke, support, elderly people, review, self-efficacy

## Abstract

A systematic review was undertaken to determine the efficacy of self-management interventions for people with stroke over the age of 65 in relation to psychosocial outcomes. PubMed, Embase, and PsycInfo were searched for randomized controlled clinical trials. Studies were eligible if the included people with stroke had a mean age ≥65 years in both the intervention and control group. Data on psychosocial measurements were extracted and an assessment of methodological quality was undertaken. Due to heterogeneity across the studies, the results were synthesized narratively. Eleven studies were identified. They included different self-management interventions in terms of theoretical rationales, delivery, and content. Seven psychosocial outcomes were identified: i) self-management, ii) self-efficacy, iii) quality of life, iv) depression, v) activities of daily living, vi) active lifestyle, and vii) other measures. Self-management interventions for people with stroke over the age of 65 may be beneficial for self-management, self-efficacy, quality of life, activity of daily living, and other psychosocial outcomes. However, low study quality and heterogeneity of interventions, as well as variation in time of follow-up and outcome measures, limit the possibility of making robust conclusions.

## 1. Introduction 

In a welfare state like Denmark, people retire when they are approximately 65 years old [1], which means that around 20% of life is lived as an old age pensioner [2]. It is often around the age of retirement that lifestyle and/or predisposed factors lead to illness, leaving the elderly people with the greatest health inequalities, both socially and biologically. Research has shown that those who thrive best in the transition from working life to retirement, and as old age pensioners in general, are those who conduct active lifestyles including elements also evident in working lifestyles [3]. These elements may disappear when a person is no longer in the labour market. Old age pensioners are at risk of experiencing loneliness due to termination of work-based social relationships [4]. In Denmark, around two-thirds—approximately 10,000 annually—of all new incidents of stroke affect people over the age of 65 [5]. The consequences of stroke often involve long-term physical, psychological, cognitive, and behavioral difficulties [6,7,8,9,10,11]. As more people survive stroke, the number of old age pensioners living with long-term consequences due to the stroke increases [12]. Elderly people with stroke are an especially vulnerable group, prone to loneliness, social isolation, and lack of social reintegration due to bodily frailty, inactivity, and difficulties in terms of social relationships after leaving the labor market.

### Self-Management

Two systematic reviews by Fryer et al. and Wray et al. have investigated the effects of self-management interventions, and found that self-management interventions may improve quality of life and hence self-efficacy post-stroke [13,14]. However, they investigated the effect of self-management interventions on the entire adult population with stroke, leaving a lack of knowledge on which core elements of self-management interventions are effective for people with stroke over the age of 65. There is no universal definition of self-management, but it is commonly described as an individual’s ability to manage the symptoms, treatment, physical and psychological consequences, and lifestyle changes inherent in living with chronic disease [15]. The American guidelines for stroke rehabilitation recommend supporting self-management through exercises [16]. Jones et al. advocate an approach that facilitates people with stroke actively reflecting on, and taking initiative and responsibility for, their activities [17]. In an on-going project, Stroke 65 plus—continued active life [18], this is supplemented by involving social networks and social contexts [19]. A similar approach is recommended in a qualitative meta-synthesis by Walsh et al. [20]. Given the growing number of elderly people with stroke, it seems relevant to investigate whether psychosocial self-management support is beneficial for elderly people following stroke.

The objective of this systematic literature review is to determine the efficacy of self-management interventions for people with stroke over the age of 65 in relation to psychosocial outcomes.

## 2. Method

The method and included studies are reported according to the Preferred Reporting Items for Systematic Reviews and Meta-Analyses (PRISMA) guidelines [21] (Appendix A). A review protocol was developed and registered in the International Prospective Register of Systematic Reviews (PROSPERO). (Submitted on 11th November 2019. Being assessed on June 16th 2020).

### 2.1. Eligibility Criteria

Study design: Randomized controlled clinical trials published in English, Danish, Norwegian, or Swedish.

Population: People with stroke over the age of 65 (mean age > 65 in both the intervention and control group), in any setting e.g., hospital, home, or community-based and at any point in time post-stroke.

Intervention: Interventions focusing on self-management were defined as interventions that were designed to facilitate behavioral change and improvements in psychological or social functioning. The intervention had to be provided by health professionals, possibly combined with technology. The intervention could be individual or group based.

Comparator: Any control condition.

Outcomes: Psychosocial measurements e.g., quality of life, self-efficacy, mental health, activity of daily living, or change to an active lifestyle. Measurements could be both primary and secondary outcomes.

Exclusion criteria: 1) Studies that investigated chronic diseases e.g., heart failure, traumatic brain injury or tumors. 2) Interventions that only supported caregivers. 3) Interventions that provided education/information, workbook/diary, medication, or exercise only.

### 2.2. Search Strategy

The search strategy for this study was developed in the light of an initial search related to the development of a self-management protocol in January 2017 [18]. The initial search involved an iterative process to find the most relevant databases and search terms. This resulted in a search for literature in three databases on 28th May 2019: PubMed (1966 to May 2019), Embase (1980 to May 2019), and PsycInfo (1967 to May 2019). A follow-up search in the three databases was added on 29th April 2020. Two more articles were included on this basis. Table 1 outlines the hits in each database for the overall search. In addition, the reference list of included studies and two systematic reviews were searched for relevant studies [13,14]. The final selection of databases and search terms was developed with the help of a research librarian experienced in advanced literature searches. The search terms were developed for PubMed and adapted for the other databases. When possible, the searches were based on specific subject headings, e.g., Medical Subject Heading (MESH terms). If there were no specific subject headings, only words in the title, abstract, and keywords were used. Apart from search terms related to stroke, the search included the term “self-management” and the synonyms “self-care”, “self-efficacy”, “patient participation”, and “autonomy”. A full list of the electronic search strategies is available in Appendix A.

### 2.3. Study Selection and Data Extraction

Studies were screened to remove duplicates using the software-based reference management system RefWorks [22]. Following the removal of duplicates, the titles and abstracts of the remaining studies were screened by the first and last author for relevance according to the aforementioned inclusion and exclusion criteria. Full-text studies were retrieved when eligibility could not be determined from the abstract alone. Throughout the process, the first author and last author discussed the relevance of a study, if in any doubt. The first author and last author also independently screened the papers for eligibility. Data extraction was undertaken by the first author and last author, and the findings are reported in Table 2, which includes citation (authors, title, publication year, and journal), country (of origin), sample size, mean age, study design, population, study aim, theoretical foundation, intervention/comparator, psychosocial outcome measures, and follow-up time. Only published data were extracted.

### 2.4. Assessment of Risk of Bias

The methodological quality of the studies was assessed by the first author and last author independently of each other using the revised Cochrane risk-of-bias tool for randomized trials (RoB2) [23]. Five headings were associated with biases in RoB2: the randomization process, deviations from intended interventions, missing outcome data, measurement of the outcome, and selection of reported results. A judgement of bias for each domain was facilitated by an algorithm that maps a proposed judgement. The response options were “yes”, “probably yes”, “probably no”, “no”, and “no information”; and the judgement of risk of bias for each domain was “low risk of bias”, “some concern”, or “high risk of bias”. Furthermore, a judgement of the overall risk of bias was made. The overall risk of bias judgement was the same as for the individual domains, and generally corresponds to the worst risk of bias in any of the domains. Quality assessment was not used to exclude studies; however, potential limitations of the research were highlighted. In case of discrepancy between the first author and last author, consensus was obtained through collaboration with the team of authors.

### 2.5. Data Synthesis

The intervention characteristics are narratively presented in categories based on the theoretical foundations. The evidence synthesis is shown in categories of the psychosocial outcomes. Differences between the intervention and control groups from baseline to follow-up were collected for the psychosocial outcomes. Estimates indicating a difference between the groups were given where possible. It was reported whether the difference between the groups was statistically significant or not. *p*-values < 0.05 were considered statistically significant. In those cases where it was not possible to report *p*-values, the confidence interval was specified.

## 3. Results

The PRISMA flow diagram [21] of the review process and selection of studies is shown in Figure 1. Once duplicates were removed, 820 references were screened for eligibility. Full text was obtained for 43 studies, 32 of which were subsequently excluded. Reasons for exclusion were not being able to obtain full text on three occasions [24,25,26], two studies turning out to be protocols [27,28], ten studies having a population with a mean age under 65 years [29,30,31,32,33,34,35,36,37,38], three study interventions being based on body functions [39,40,41], three studies only including outcome measures at body level [42,43,44], two studies not including the target population or including other chronic diseases [45,46], three studies focusing on education or exercise only [47,48,49], two studies providing peer or caregiver support [50,51], three studies involving a workbook or diary as the only content in the intervention [52,53,54], and one study being based on art [55]. This resulted in eleven studies eligible for inclusion in this review [56,57,58,59,60,61,62,63,64,65,66]. The reference lists of these eleven studies were hand-searched for additional studies that met the inclusion criteria. No additional studies were identified. Data from these eleven articles were extracted for synthesis.

### 3.1. Study Characteristics

In total, 2216 participants were included; the mean age ranged from 66.0 [62,63] to 71.7 years [59] in the intervention groups, and from 65.0 [63] to 73.0 years [59] in the control groups. None of the studies focused only on elderly people with stroke over the age of 65. Two studies only mentioned a total mean age for both the intervention and control groups of 67.5 and 70.1 years, respectively [58,65]. Sample sizes ranged from 40 [62] to 400 [59]. Two studies were either a pilot or feasibility study [58,62]. The eleven included studies evaluating self-management interventions included different strategies to enhance quality of life, activities of daily living, change of lifestyle, and participation in society. The different interventions were provided by health professionals, mostly by nurses, occupational therapists, or physiotherapists. The duration of the interventions varied from 6 weeks to 21 months and included 1 to 16 meetings. In one study, only telephone calls were made, and no face-to-face meetings were conducted [58]. Three studies included people with stroke as well as their caregivers [58,60,62], and in one study family members or friends could be present at the participant’s request [59]. More than 55 different outcome measurements were used to assess the interventions, 33 of which were considered to evaluate psychosocial functioning. An overview of the 33 outcome measurements is available in Appendix A. Eighteen of the psychosocial measurements were primary outcome measures. However, six out of the eleven studies did not differentiate between primary and secondary outcome measures. The characteristics of the included studies are shown in Table 2.

### 3.2. Intervention Characteristics

The description of the theoretical foundations varied. However, they could be categorized into at least three main themes: i) psychosocial theories of stress, coping, empowerment, self-efficacy, and self-determination [58,59,60,63,64,65,66], ii) the Trans-theoretical Stages-of-Change and motivational interviewing [61], and iii) various not clearly defined ideas of health, prevention, well-being, and client-centered self-care, which was supposed to increase functioning and prevent post-stroke complications—in the present study this is described as health literacy [56,57,62]. In these studies, no clear theoretical rationales were defined. Instead, the authors referred to results associated with better stroke recovery found in other studies. Furthermore, the two studies examining a family intervention designed to influence social support and show that self-efficacy affects functional outcome, as well as improving quality of life for caregivers, lacked a description of the theoretical foundations of the intervention [58,60]. The most recent studies were better theoretically grounded and had a broader biopsychosocial understanding and application of this understanding in their self-management interventions. All of the included studies carried out interventions meeting the criteria of self-management, though in quite varying forms.

The interventions grounded in psychosocial theories were diverse in delivery and content [58,59,60,63,64,65,66]. The family intervention assisting people with stroke and their caregivers during transition to home identified problems in five key areas: (1) family functioning, (2) mood, (3) neurocognitive functioning, (4) functional independence, and (5) physical health. This was followed up by telephone calls [58]. Another intervention was conducted as group education encouraging the participants to (1) engage in activities that promote health and wellbeing, such as adopting healthy behaviors (e.g., exercise and healthy eating), (2) minimize the negative influence of their illness on their lives, (3) manage the negative emotional impact of their symptoms, and (4) take an active role in their own health by developing partnerships with health professionals [64]. A nurse-led self-efficacy-based stroke self-management program of four weeks used Bandura constructs of self-efficacy and outcome expectation [65]. The intervention included a home visit (week 1), two hours of community group sessions (week 2–3), and three follow-up phone calls (week 4) [65]. Sit et al.’s study from 2016 described a 13-week stroke patient empowerment intervention (Health Empowerment Intervention for Stroke Self-management) [66]. A nurse facilitator provided feedback using self-management steps and problem-solving strategies to strengthen confidence and motivation. The intervention consisted of six weekly small-group sessions from week three to week eight in parallel with the ambulatory rehabilitation schedule (usual care), and included the home-based implementation during weeks 9 to 13 with biweekly telephone follow-up calls to the participants during this period [66]. A novel, community-based self-directed rehabilitation intervention “Take Charge” was described in the study by Fu et al. The intervention consisted of one-to-one, non-directive exploration of the participants views on what and who was important to them in their lives, and what they wanted to prioritize for the next 12 months. The participants received one or two sessions [59]. A dialogue-based intervention which consisted of eight individual sessions addressing feelings, thoughts, and reflection related to the participants’ experiences after stroke was presented in the study by Hjelle et al. [63].

The Trans-theoretical Stages-of-Change model and motivational interviewing (MI), to frame the intervention strategy, was applied by Green et al. [61]. The Stages-of-Change model consists of a six-level scale in which clinicians assess readiness to change in a cyclical manner. Green et al. [61] described a group educational intervention involving nurse-mediated motivational counselling and scaled lifestyle classes. The educational material included information about causation, consequences, stroke recovery, and available community resources. Using MI techniques, the study nurse and each individual participant identified a health behavior needing change and identified barriers and facilitators to overcome these behaviors [61].

Within the category of health literacy, two of the studies were identified as post-stroke care management interventions, and focused on medical issues, such as hypertension and smoking cessation and their possible treatment [56,57]. One of these studies provided lifestyle and behavioral strategies to patients and caregivers such as a low-sodium diet, weight reduction techniques, education materials, and instructions to stop smoking [56]. This intervention was further developed as a comprehensive post-discharge care management [57]. It also included home visits to ensure that needed social services (e.g., Meals on Wheels, pre-packaged medication systems, and home health aides) were in place to maximize quality of life. Additionally, frequent assessments and interventions to reduce common post-stroke complications (e.g., depression, incontinence, and falls) were conducted. Guidetti and Ytterberg developed and applied a client-centered self-care intervention that comprised nine steps [62]. The occupational therapist established a relationship with the stroke client, and together they identified the client’s difficulties in performing meaningful activities, and set goals using the Canadian Occupational Performance Measure; afterwards, they planned strategies for self-care. The stroke client was introduced to a training diary in order to resume responsibility for his/her own training and goals. The caregiver was invited to collaborate. The intervention included strategies to increase awareness of ability/disability, find new ways (compensation) of performing self-care, and/or modify environmental demands to enable performance [62].

### 3.3. Risk of Bias

An estimation of possible risk of bias is presented in Figure 2. None of the eleven studies were judged to have low risk of bias across all domains. Seven of the studies had a high risk of bias in at least one domain, which is why the overall risk of bias was judged also to be high [56,60,61,62,64,65,66]. The overall risk of bias in the remaining four studies raised some concern [57,58,59,63]. The major methodological weakness of the included studies was bias in measurement of the outcome as none of the outcome assessors were blinded to group assignment. Three studies also seemed to use investor-generated non-standardized measurements which were unlikely to be sensitive to plausible intervention effects, and/or were likely to have poor validity [57,60,61]. Furthermore, missing outcome data due to loss of follow-up (>20%) and missing/insufficient methods correcting for missing outcome data, was a methodological weakness. None or insufficient details limited the ability to determine whether data in the included studies were analyzed in accordance with a pre-specified plan in nine of the studies. An overview of the answers to signaling questions, together with free-text justification of the answers, is available in Appendix A.

### 3.4. Evidence Synthesis

Due to the identified heterogeneity of the interventions combined with a lack of comparable outcome measures and/or different times of follow-up, it was considered inappropriate to pool data and conduct a meta-analysis across the studies. The efficacy of the interventions is therefore synthesized narratively.

### 3.5. Efficacy of Self-Management Interventions

The efficacy of self-management interventions for people with stroke over the age of 65 in relation to psychosocial outcomes is presented below according to the following outcome measures: i) self-management, ii) self-efficacy, iii) quality of life, iv) depression, v) activity of daily living, vi) active lifestyle, and vii) other measures.

### 3.6. Self-Management

Self-management was an outcome measure in two recent studies [65,66]. Both studies showed a significant improvement in the intervention group within different areas, e.g., cognitive symptom management, outcome expectation, and satisfaction with performance of self-management behaviors. Lo et al. measured both expectation and satisfaction with performance of self-management behaviors, using two kinds of measurements, the Stroke Self-Management Outcome Expectation Scale and the Stroke Self-Management Behaviors Performance Scale. At follow-up at eight weeks, the intervention group showed significant improvements in both outcomes (mean difference 9.74, *p* < 0.01 and 8.63, *p* < 0.01, respectively). This was consistent with Sit et al. who used the Chinese Self-Management Behavior Questionnaire. This questionnaire was divided into four areas in which cognitive symptom management (six items) and communication with physicians (three items) were included. They found significantly better cognitive symptom management behavior at one week, three months, and six months of follow-up in the intervention group (all *p* < 0.001). Furthermore, the intervention group had significantly better communication with their physicians at one week (*p* < 0.001) and three months of follow-up (*p* = 0.002). The difference between the groups regarding communication with the physician was not significant at six months of follow-up (*p* = 0.094). In both studies, the interventions seemed to facilitate self-management.

### 3.7. Self-Efficacy

Self-efficacy was used as an outcome measure in four studies [60,64,65,66]. Three of them showed improvements in self-efficacy in the intervention groups, suggesting that self-management interventions may have a positive effect on self-efficacy. In the study by Kendall et al., the authors measured self-efficacy by the Self-efficacy Scale at three, six, nine, and twelve months. The control group demonstrated lower levels of self-efficacy than the intervention group, and self-efficacy in the control group was consistently lower than in that of the intervention group. The difference between the groups was significant in self-efficacy levels across all times of follow-up (*p* = 0.003). Sit et al. measured self-efficacy in illness management at one week, and at three and six months. They used the Chinese Self-Management Behaviour Questionnaire, six items of which relate to self-efficacy. Participants in the intervention group showed significantly better self-efficacy in illness management at the three-month and six-month follow-up (mean difference 7.3, *p* = 0.011 and 7.5, *p* = 0.012, respectively). Self-efficacy was also measured by Lo et al. using the Stroke Self-Efficacy Questionnaire. They reported that the intervention group had significant improvements in self-efficacy at eight weeks of follow-up (mean difference 7.50, *p* < 0.01). Glass et al. created for their study 10 questions to assess participants’ recovery self-efficacy and found no effect of the intervention (*p* = 0.97).

### 3.8. Quality of Life

Of the eleven studies, six measured quality of life [56,57,59,60,63,64]. To some degree, they showed an effect of the self-management interventions. Allen et al. were confident that their intervention positively affected quality of life in the intervention group compared with usual care at three months of follow-up (effect size 0.52, 90% CI = 0.12 to 0.91). They used the Stroke Adapted 30-item Sickness Impact Profile to measure quality of life. Fu et al. used the Short Form 36 Physical Component Summary (SF-36 PCS), Short Form 12 Physical Component Summary (SF-12 PCS), and European Quality of Life-5 Dimensions-5 levels (EQ-5D-5L) to measure health-related quality of life at either six or twelve months. SF-12 PCS was significant at six months (mean difference 2.4, *p* = 0.02) and SF-36 PCS was significant at twelve months (mean difference 1.8, *p* = 0.02), both in favor of the two intervention groups. EQ-5D-5L showed no significant difference between the groups at twelve months after the stroke (mean difference 2.9, *p* = 0.21). They concluded that two sessions, six weeks apart, were better than a single session. Kendall et al. used the Stroke-Specific Quality of Life scale (SSQOL). In the SSQOL, quality of life is divided into 12 domains: five physical domains (energy, language, vision, mobility, and fine motor tasks), three psychological domains (mood, personality, and thinking), three social domains (social roles, family roles, and work productivity), and self-care. The study showed a significant impact of the intervention on the quality of family roles (*p* < 0.01) and a trend towards significance in relation to self-care (*p* = 0.05), work productivity (*p* < 0.05), and functioning in daily activities (*p* < 0.05) at nine months post-stroke. The difference between the two groups disappeared at 12 months post-stroke. Allen et al. also used the SSQOL and found no difference between the intervention and control group at the six-month follow-up (mean difference = 2, 95% CI = −9.0, 5.0). They concluded that both groups had relatively good quality of life at six months. Furthermore, no significant differences were found between the groups in the study by Hjelle et al. at six months (mean difference 0.03, *p* = 0.64). They used the Stroke and Aphasia Quality of Life Scale-39. Glass et al. stated that they would measure quality of life (and self-rated health) using a five-level, single-item global rating scale. However, they chose not to report the results in the article.

### 3.9. Depression

Five studies evaluated depression, and none of them showed an effect of the self-management interventions [56,57,58,60,63]. In three studies, depression was measured using the Center for Epidemiologic Studies Depression Scale (CES-D). The CES-D scores in the study by Allen et al. implied that the participants with greater baseline deficits obtained the greatest relative benefits from the intervention. However, the effect was not significant between the groups at three months of follow-up (mean difference −0.23, 90% CI = −1.4, 0.69). Very little time was devoted to addressing allied health (or psychosocial) issues in the study by Allen et al. However, a subgroup analysis of the CES-D was made. No effect was identified when measuring the mean difference between the groups at six months of follow-up (0.2, 95% CI: −0.2, 0.8). In the study by Glass et al., the results suggested that their intervention was more effective in patients with better mental health. Regarding depression, the intervention showed no effect (*p* = 0.75) at six months of follow-up. Glass et al. explained this in terms of group composition, as fewer participants in the control group were depressed. In the study by Hjelle et al., a difference between the groups regarding depression at baseline was also seen, as fewer participants in the intervention group were depressed. After controlling for baseline depression, no significant difference between the groups was shown at six months follow-up (mean difference 1.25, *p* = 0.51). Bishop et al. used the 13-item Geriatric Depression Scale Short Form to measure depression. They found no significant difference in levels of depression between the groups at three and six-months of follow-up (*p* > 0.10).

### 3.10. Activities of Daily Living

Two studies had daily activities as outcome measures. They showed some effect of the self-management interventions [62,66]. Sit et al. used the Chinese Lawton Instrumental Activities of Daily Living Scale to assess the participants’ ability to perform tasks such as using a telephone, doing laundry, and handling finances. They reported a clear, significant improvement in the intervention group at one week, three months, and six months of follow-up (all *p* < 0.001). Guidetti and Ytterberg used the Stroke Impact Scale, subscale 5 to access the perceived difficulties in typical activities of daily living at 12-months of follow-up, e.g., cutting food with a knife and fork, getting to the toilet in time, and going shopping. They showed no significant difference between the groups (*p* = 0.55). Furthermore, Guidetti and Ytterberg measured the occupational gaps between what the participants wanted to do and what they actually did using the Occupational Gaps Questionnaire, showing a tendency of significant difference between the groups at 12 months of follow-up (*p* = 0.08).

### 3.11. Active Lifestyle

Physical activity—here understood as an active lifestyle—was measured in three studies [57,60,61]. Results were reported in two of the studies and showed no effect of the self-management interventions. Allen et al. investigated knowledge of stroke risk factors including present exercise using an investigator-generated questionnaire measuring stroke knowledge and lifestyle modification in people with stroke. No effect was found between the groups at six months of follow-up (mean difference 10.0 95%, CI: −0.1 to 20). Green et al. undertook a secondary analysis based on a description of lifestyle risk factors identified by the individual participant. Exercise was defined as a lifestyle risk factor by 134 of 197 participants. Of the 134 participants, 67 chose to work with exercise as a goal. At three months of follow-up, 72.2% of the 67 participants had changed from a passive to a physically active lifestyle (*p* = 0.000). However, no significant difference was seen between the intervention and control group (*p* = 1.00). Physical performance was measured at baseline in the study by Glass et al. by combining the scores of five timed tests of functional capacity, e.g., writing a sentence and walking 20 feet. However, the results were only used to construct a chronic disease score with other health-related variables.

### 3.12. Other Measures

In addition to the above subdivision of the outcome measures, ten other psychosocial measures were used in six different studies [58,59,60,61,62,63]. The results are summarized below.

Frequency of activities was measured by Bishop et al. using the Frenchay Activities Index and showed a significant improvement regarding the stroke caregivers in the intervention group at three months follow-up (*p* < 0.05). The effect continued as a trend at six months of follow-up (*p* < 0.10). In the study by Fu et al., they also showed a significant improvement favoring the intervention groups at twelve months (mean difference 2.7, *p* = 0.01) using the Frenchay Activities Index. Guidetti and Ytterberg also used the Frenchay Activities Index. However, they showed no significant difference between the intervention and control group at 12 months of follow-up (*p* = 0.54). Guidetti and Ytterberg measured the participants’ perceived difficulties in participation using the Stroke Impact Scale, subscale eight, such as work, the role as a family member, and the ability to control life as desired. Their study showed a trend of significant difference between the groups at 12 months of follow-up (*p* = 0.09). Finally, Guidetti and Ytterberg reported no significant difference in perceived satisfaction with life using the Life Satisfaction Scale between the intervention group and the control group (*p* = 0.62) at 12 months of follow-up. In the study by Green et al., stress was defined as a lifestyle risk factor. However, it was not reported how many of the participants also considered stress as a lifestyle risk factor after a stroke, nor were any possible differences or changes between the groups reported. Received social support was investigated by Glass et al. They found no significant difference between the groups at six months of follow-up (*p* = 0.26) [60]. To measure psychosocial well-being Hjelle et al. used the General Health Questionaire-28. No significant difference between the groups was shown at six months follow-up (mean difference 0.30, *p* = 0.80). Furthermore, Hjelle et al. measured sense of coherence with Sense of Coherence Scale-13. No difference between the groups was found either (mean difference 0.28, *p* = 0.73). Fu et al. use the Caregiver Strain Index in their study. No significant difference between the groups were showed (mean difference 0.10, *p* = 0.78). Bishop et al. measured family functioning using the Family Assessment Device (FAD) and the Perceived Criticism Scale. They found a positive effect on family functioning, which became more pronounced as the intervention progressed. They found that a significant group difference favored the intervention group at six months of follow-up measured by the FAD (*p* < 0.05). In addition, they revealed a significant difference in perceived criticism of self by family for the person with stroke and caregiver dyad as a whole, and for the caregivers individually at six months of follow-up (*p* < 0.05).

## 4. Discussion

The objective of this systematic review was to determine the efficacy of self-management interventions for people with stroke over the age of 65 in relation to psychosocial outcomes. The review identified eleven randomized controlled trials involving 2216 participants with stroke. The mean age ranged from 66.0 to 71.7 years in the intervention groups and from 65.0 to 73.0 years in the control groups. None of the studies focused only on elderly people over the age of 65. Two studies were either a pilot or a feasibility study. Due to the heterogeneity of the interventions identified (with respect to theoretical rationales, delivery, and content) and lack of comparable outcome measures, it was considered inappropriate to pool data and conduct a meta-analysis across the studies. The efficacy of the interventions was synthesized narratively. Thirty-two outcome measures were considered to evaluate psychosocial functioning, reflecting the following seven psychosocial domains: i) self-management, ii) self-efficacy, iii) quality of life, iv) depression, v) activity of daily living, vi) active lifestyle, and vii) other measures.

### 4.1. Methodological Quality of Included Studies

The methodological quality of the included studies was low. In seven of the eleven trials, the overall risk of bias was judged to be high. Methodological weaknesses differed across the studies. However, bias due to the “measurement of the outcome” domain of RoB2 was seen in all studies. According to RoB2, the outcome assessor is the participant when the outcome is participant-reported outcomes [67]. In this light, none of the outcome assessors were blinded, as it was impossible to blind the participants to group assignment in the studies. Thus, outcome assessment is potentially influenced by knowledge of the intervention, which might increase performance and ascertainment bias. The three studies measuring physical activity seemed to use investigator-generated, non-standardized measurements, which might affect data quality as it is uncertain whether they can be used to capture any effect [57,60,61]. Furthermore, bias due to the “missing outcome data” domain of RoB2 was observed in eight of the eleven studies. This is problematic as the participants missing from the analysis may vary systematically from those who were included [68], and the values of the missing outcome data could make an important difference to the estimated intervention effect [23]. The results of the studies identified as being at high risk of bias should be interpreted with caution.

### 4.2. Limitations of the Review

This review focuses on elderly people with stroke over the age of 65. However, none of the included studies focused only on elderly people over the age of 65. Therefore, the interventions were not directly targeted at this age group. This is a limitation of the present review. The review aimed to be as inclusive as possible of potential self-management interventions. However, it may be criticized for being overly inclusive of interventions that did not identify themselves as “self-management” or defined the interventions as facilitating behavioral change or improvements in psychological or social functioning. For example, the two studies by Allen et al. (2002) and Allen et al. (2009) used the term “care management” [56,57], which might be a mix between self-care and self-management. Another example is the lack or insufficient description of theoretical foundations [56,57,58,60,62,64]. Furthermore, the definition of psychosocial outcomes is not unambiguous. The authors of this paper identified theoretical foundations and psychosocial outcomes based on knowledge of health behavior theories as well as their expertise in the public health field, which may result in potential biases. Since a narrative synthesis was conducted rather than a meta-analysis in this review, other study designs could have been included to identify additional dimensions and strengthen the evidence of “what works” in self-management interventions to elderly people over the age of 65.

### 4.3. Interpretation and Implications for Future Research

In a recent study, self-management is described as a multidimensional concept, which suggests that chronic disease self-management should be understood as a “fluid, iterative process during which patients incorporate multidimensional strategies that meet their self-identified needs to cope with chronic disease within the context of their daily living” [69]. Essentially, self-management is a process that affects and leads to outcomes rather than being an end point or outcome in and of itself [70,71,72]. However, Lo et al. and Sit et al. used self-management as an outcome in their studies [65,66]. Researchers should be aware of this divergence in process and outcome-oriented thinking regarding self-management in future research.

All the interventions in the included studies could be regarded complex [73]. According to the Medical Research Council (MRC) Framework, the development of a complex intervention should be based on a four-stage process of “develop-test-evaluate-implement” [73]. Only one study used the MRC Framework to develop and feasibility test their intervention before implementing into their randomized controlled trial [63]. Future research in complex interventions should be designed and feasibility tested with implementation in mind at the very start as it is likely to improve the chances of embedding the new intervention into routine practice.

Physical activity is an important part of a healthy lifestyle and is crucial for secondary prevention as it improves physical and mental functioning, especially in the elderly population [74,75]. Physical activity is also important from a biopsychosocial perspective since it reduces depression [76]. No beneficial effect of self-management interventions on physical activity could be found in the reviewed literature. Though health-promoting effects of physical activity are part of many national guidelines from health authorities, the difficulty of affecting behavioral changes in this field is widely acknowledged [77]. Future research should include objective measures of physical activity, such as accelerometry.

Only one study measured participation, which is striking since relations to family and friends and participation in society are crucial for well-being after a stroke [78]. In future research an increased focus on involving relatives and network in stroke self-management interventions is recommended.

Overall, this study reveals the need to improve understandings of how to affect change in self-management interventions for people with stroke over the age of 65. Development of more effective self-management interventions for elderly people with stroke based on a solid theoretical foundation and scientific methods is recommended. In addition, greater awareness of linking behavior change theory to designing and evaluating self-management interventions for change behavior among elderly people with stroke is required.

## 5. Conclusions

Self-management interventions for people with stroke over the age of 65 might be beneficial for self-management, self-efficacy, quality of life, activities of daily living, and other psychosocial outcomes. However, low study quality and heterogeneous interventions, as well as variation in time of follow-up and outcome measures, limit the possibility of making robust conclusions. The development of more effective self-management interventions for elderly people with stroke based on solid theoretical foundations and scientific methods is recommended.

## Figures and Tables

**Figure 1 geriatrics-05-00038-f001:**
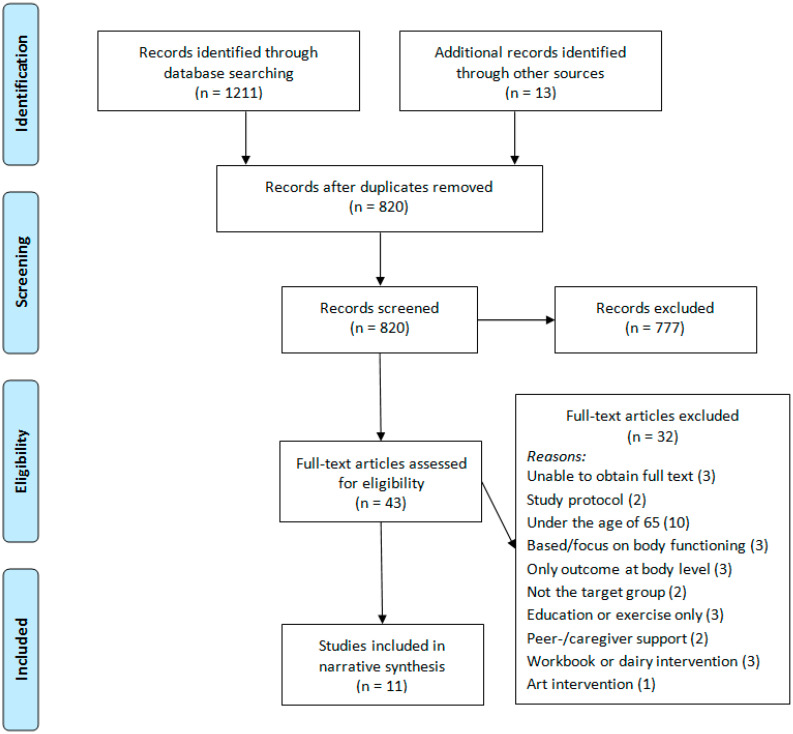
PRISMA flow diagram of study selection.

**Figure 2 geriatrics-05-00038-f002:**
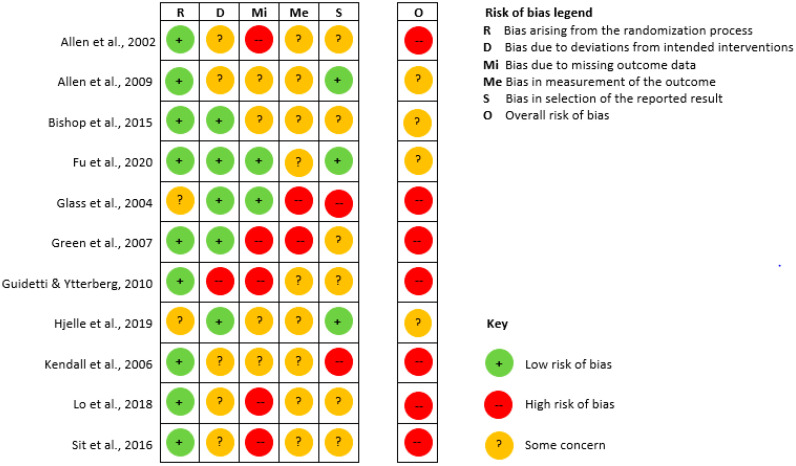
Risk of bias.

**Table 1 geriatrics-05-00038-t001:** Database search information.

Database	Fields Searched	Articles Identified (Hits)
PubMed	Title, Abstract and Keywords	488
Embase	Title, Abstract and Keywords	603
PsycInfo	Title, Abstract and Keywords	120
Total (*n*)		1211

**Table 2 geriatrics-05-00038-t002:** Characteristics of included studies.

Authors, Title, Year, Journal, Country	Sample Size	Age (Mean and SD)	Study Design	Population	Aim	Theoretical Foundation	Intervention/ Comparator	Psychosocial Outcome Measures	Follow-up Time Point
Allen, K.R.; Hazelett, S.; Jarjoura, D.; Wickstrom, G.C.; Hua, K.; Weinhardt; J., and Wright K. Effectiveness of a Post discharge Care Management Model for Stroke and Transient Ischemic Attack: A Randomized Trial 2002 Journal of Stroke and Cerebrovascular Diseases USA [56]	96	Intervention group 69.0 (SD not reported) Control group 72.0 (SD not reported)	RCT- Two arms	Ischemic stroke and transient ischemic attack	To test the effectiveness of comprehensive, interdisciplinary post-discharge care management for improvement of a profile of indicators of health recovery and secondary prevention in stroke and transient ischemic attack (TIA) patients.	Health literacy, but no clear theoretical rationales were defined.	Intervention: An advanced practice nurse-care manager provided care management focused on health promotion and psychosocial well-being. Comparator: Usual care	1) QOL1 2) Depression	Three months post-discharge.
Allen, K.R.; Hazelett, S.; Jarjoura, D.; Hua, K.; Wright, K.; Weinhardt, J., and Kropp D.A. Randomized Trial Testing the Superiority of a Post discharge Care Management Model for Stroke Survivors 2009 Journal of Stroke and Cerebrovascular Diseases USA [57]	380	Intervention group 68.0 (SD not reported) Control group 69.0 (SD not reported)	RCT- Two arms	Ischemic stroke	To test the superiority of comprehensive interdisciplinary post-discharge stroke care management for improving outcomes for stroke survivors as compared with organized acute stroke department care with enhanced discharge planning.	Health literacy, but no clear theoretical rationales were defined.	Intervention: An advanced practice nurse-care manager provided care management including self-management support. Comparator: Usual care	1) QOL1 2) Depression 3) Active Lifestyle	Six months post-discharge.
Bishop, D.; Miller, I.; Weiner, D.; Guilmette, T.; Mukand, J.; Feldmann, E.; Keitner, G., and Springate, B. Family Intervention: Telephone Tracking (FITT): A Pilot Stroke Outcome Study 2015 Topics in Stroke Rehabilitation USA [58]	49	Both groups Stroke individuals: 70.1 (SD 11.6) Caregivers: 56.8 (SD 16.4) Only a total mean age for both groups is reported	RCT- Two arms	Stroke and their caregivers (Not sub-arachnoid hemorrhage)	To preliminarily test the efficacy of a telephone intervention.	Grounded in psychosocial theories (A family system approach).	Intervention: A family intervention by telephone tracking designed to assist people with stroke and their primary caregivers during the first 6 months after stroke. Comparator: Usual care	1) Depression 2) Functional independence2 3) Family functioning2	Three- and six-months post-stroke.
Fu, V.; Weatherall, M.; McPherson, K.; Taylor, W.; McRae, A.; Thomson, T.; Gommans, J.; Green, G.; Harwood, M.; Ranta, A.; Hanger, C.; Riley., and McNaughton, H. Taking Charge after Stroke: A randomized controlled trial of a person-centered, self-directed rehabilitation intervention 2020 International Journal of Stroke New Zealand [59]	400	Intervention groups Take Charge 1: 71.4 (SD 12.6) Take Charge 2: 71.7 (SD 12.6) Control group 73.0 (SD 12.2)	RCT- Three arms	Stroke	To confirm whether the Take Charge intervention improved quality of life at 12 months after stroke and whether two sessions were more effective than one.	Grounded in psychosocial theories (Self Determination Theory).	Intervention: Take Charge: A one-to-one, non-directive exploration of the stroke individuals views on what and who was important to them in their lives, and what they wanted to prioritize for the next 12 months Take Charge 1: A single Take Charge session. Take Charge 2: Two Take Charge sessions six weeks apart. Comparator: Were given written educational material about stroke covering common issues following stroke and risk factor management.	1) QOL1 2) Actual activities2 3) Caregiver strain2	Six- and 12-months post-stroke.
Glass, T.A.; Berkman, L.F.; Hiltunen, E.F.; Furie, K.; Glymour, M.; Fay, M.E., and Ware, J. The Families in Recovery from Stroke Trial (FIRST): Primary Study 2004 Psychosomatic Medicine USA [60]	291	Intervention group 69.3 (SD 11.0) Control group 70.4 (SD 11.0)	RCT- Two arms	Ischemic or non-traumatic hemorrhagic stroke	To examine whether a family-system intervention designed to influence social support and self-efficacy affects functional outcome in older stroke patients.	Grounded in psychosocial theories (A family system approach).	Intervention: An integrative psychosocial intervention for stroke individuals and their families tailored to each family’s needs Comparator: Usual care	1) Self-efficacy 2) QOL1 3) Depression 4) Active lifestyle 5) Social support2	Three and six months post-randomization.
Green, T.; Harley, E.; Eliasziw, M., and Hoyte, K. Education in stroke prevention: Efficacy of an educational counselling intervention to increase knowledge in stroke survivors 2007 Canadian Journal of Neuroscience Nursing Canada [61]	200	Intervention group 66.3 (SD 12.4) Control group 67.2 (SD 12.4)	RCT- Two arms	Stroke and transient ischemic attack	To examine the impact of one-to-one brief nurse/patient interview on acquisition of knowledge of stroke and influence on lifestyle behavior changes.	Trans-theoretical stages of change model.	Intervention: An education-counselling interview, where participants mapped their individual risk factors on a stages-of-change model and received an appointment to the next group lifestyle class. Comparator: Usual care	1) Active lifestyle 2) Stress2	Three months post-appointment.
Guidetti, S. and Ytterberg, C. A randomised controlled trial of a client-centred self-care intervention after stroke: a longitudinal pilot study 2010 Disability and Rehabilitation Sweden [62]	40	Intervention group Stroke individuals: 66.0 (SD 14.0) Caregivers: 64.0 (SD not reported) Control group Stroke individuals: 69.0 (SD 15.0) Caregiver: 63.0 (SD not reported)	RCT- Two arms	Stroke and their caregivers	To study (i) the feasibility of the study design, (ii) effects up to 12 months on activities of daily living, use of informal care and home help services, and caregiver burden.	Health literacy, but no clear theoretical rationales were defined.	Intervention: A new client-centered self-care intervention after stroke focusing on learning to use and implement a global problem-solving strategy, goal-plan-do-check when performing self-care activities. Caregivers were invited to collaborate. Comparator: Usual care	1) ADL3 2) Social/ Lifestyle activities2 3) Participation2 4) Satisfaction with life2	Three, six and 12 months post-intervention.
Hjelle, E.; Bragsted, L.K.; Kirkevold, M.; Zucknivk, M.; Bronken, B.A.; Martinsen, R.; Kvigne, K.J.; Kitzmüller, G.; Mangset, M.; Thommessen, B., and Sveen, U. Effect of a dialogue-based intervention on psychosocial wellbeing 6 months after stroke in Norway: a randomized controlled trial 2019 Journal of Rehabilitation Medicine Norway [63]	322	Intervention group 66.0 (SD 12.1) Control group 65.0 (SD 13.3)	RCT- Two arms	Stroke	To evaluate the effect of a dialogue-based intervention in addition to usual care on psychosocial well-being 6 months after stroke.	Grounded in psychosocial theories (Salutogenesis, sense of coherence, narrative theory, and ideas from guided self-determination).	Intervention: A dialogue-based intervention that aimed to support the coping and life skills of stroke. Comparator: Usual care	1) QOL1 2) Depression 3) Well-being2 4) Sense of coherence2	Six months post-stroke.
Kendall, E.; Catalano, T.; Kuipers, P.; Posner N, Buys, N., and Charker, J. Recovery following stroke: the role of self-management education 2007 Social Science & Medicine Australia [64]	100	Intervention group 66.4 (SD 15.3) Control group 66.4 (SD 14.9)	RCT- Two arms	Stroke	To examine the utility of the Chronic Disease Self-Management course as a way of promoting progressive psychosocial recovery pathways among people with stroke.	Grounded in psychosocial theories (Lazarus and Folkman’s theory of stress and coping)	Intervention: An existing self-management intervention, the Chronic Disease Self-Management Course, was used to operationalize the concept of psychosocial skill expansion. Comparator: Usual care	1) Self-efficacy 2) QOL1	Three, six, nine- and 12-months post-stroke.
Lo, S.H.S.; Chang, A.M., and Chau, J.P.C. Stroke Self-Management Support Improves Survivors’ Self-Efficacy and Outcome Expectation of Self-Management Behaviours 2018 Stroke Australia [65]	128	Both groups 67.5 (SD 11.95) Only a total mean age for both groups is reported	RCT- Two arms	Stroke	To determine the effectiveness of a new nurse-led self-efficacy-based stroke self-management program.	Grounded in psychosocial theories (Bandura construct of self-efficacy)	Intervention: A nurse-led intervention facilitating stroke self-management. Comparator: Usual care	1) Self-management 2) Self-efficacy	Eight weeks after randomization.
Sit, J.W.; Chair, S.Y.; Choi, K.C.; Chan, C.W.; Lee, D.T.; Chan, A.W.; Cheung, J.L.; Tang, S.W.; Chan, P.S., and Taylor-Piliae, R.E. Do empowered stroke patients perform better at self-management and functional recovery after a stroke? A randomized controlled trial 2016 Clinical Interventions in Aging Hong Kong [66]	210	Intervention group 67.8 (SD 14.2) Control group 70.7 (SD 13.9)	RCT- Two arms	Stroke	To examine the effects of the empowerment intervention on stroke patients’ self-efficacy, self-management behavior, and functional recovery.	Grounded in psychosocial theories (Shearer’s theory of health empowerment)	Intervention: An intervention to empower stroke individuals with “how to” knowledge and skills to enhance self-management in conjunction with their post-stroke rehabilitation journey. Comparator: Usual care	1) Self-management 2) Self-efficacy 3) ADL3	One week, three and six months post-intervention

^1^ QOL = Quality of Life; ^2^ Included in the category ‘Other Measures’, which are ten different measurements grouped together to increase readability; ^3^ ADL = Activities of Daily Living.

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
