# Peer review of "Effect of Self-Management Support for Elderly People Post-Stroke: A Systematic Review"

_geriatrics, 2020, doi:10.3390/geriatrics5020038_

Round 1

Reviewer 1 Report

The Authors present the results of a systematic review that aimed to determine the efficacy of self-management interventions for people with stroke over the age of 65 in relation to psychosocial outcomes. A total of 11 studies were included, but due to substantial heterogeneity in these studies no data pooling and meta-analyses were performed and results were only described as narratives.

Based on their findings the Authors conclude that self-management interventions for people with stroke over the age of 65 might be beneficial for several important outcomes. Still, low study quality, heterogeneous interventions, variation in time of follow-up and outcome measures limit the possibility of making any robust conclusions based on this review. The Authors also recommend that more effective self-management interventions for elderly people with stroke should be developed that are based on solid theoretical foundations and scientific methods.

The paper provides a nice overview of the current literature of a clinically relevant topic. It also clearly highlights the need for better studies in this field.

The risk of stroke increases with age and stroke incidence in many countries will likely increase due to ageing populations. Effective interventions to improve psychosocial outcome might thus have a huge impact on the well being of stroke survivors and also society as a whole.

Some comments to the paper:

1) The Introduction is rather long and general. It should be shortened and  focus more on stroke in elderly (aged >65 years) patients and impact on psychosocial outcomes. 

2) The Results sections is also rather lengthy, but otherwise well-structured and provides a good overview of the included studies and a critical appraisal using Cochrane methodology. 

3) The Discussion section should also be shortened, especially the subsections on "limitations" and "implications for future research".

Author Response

Dear reviewer 1

Thanks for your comments to improve our manuscript. We have made the corrections below according to your comments.

Comment 1) The Introduction is rather long and general. It should be shortened and focus more on stroke in elderly (aged >65 years) patients and impact on psychosocial outcomes.

We agree that the introduction is too long. We have shortened the Introduction by omitting sections that are not as relevant to people with stroke over the age of 65. In addition, we have merged sections 1 and 1.1. Our changes:

Line 41-42: Today, the state of health is far better among the elderly than it was decades ago, as living conditions and options for treatment have improved significantly. Sentence has been removed

Line 43-45: However, some continue working until they are well into their 70s. The remaining life of a 65-year-old man in Denmark is 14 years, and for a woman it is 18 years, Text has been removed

Line 45-47: A reasonably homogeneous health state is observed in the population until the age of 60 after which a more heterogeneous development in health begins. Sentence has been removed

Line 49-50: Older people without disabilities who move into a smaller home and less physically demanding environments may experience loss of activity and thus loss of function or illness [3]. Sentence has been removed

Line 53-54: such as being part of a social community, working towards the same goals and meeting needs for activity and participation in everyday life. Sentence has been removed

Line 57: Stroke Subheading has been removed

Line 58-60: Although the incidence of stroke in high-income countries is decreasing [6], stroke remains the leading cause of disability [7]. Since more people survive stroke, the number of people living with its long-term consequences increases [8]. Two sentences have been removed

Line 60-61: around 93,000 people live with post-stroke consequences – equivalent to 1.9% of the population. Text has been removed

Line 63-64: which can cause an unexpected interruption in normal life cycles Text has been removed

Line 64-68: Inserted text As more people survive stroke the number of old age pensioners living with long-term consequences due to the stroke increases [8]. Elderly people with stroke are an especially vulnerable group, prone to loneliness, social isolation and lack of social reintegration due to bodily frailty, inactivity and difficulties in terms of social relationships after leaving the labour market.

Line 68-69: Consequently, suffering stroke leaves many people more vulnerable in terms of social reintegration than they were before stroke hit Sentence has been removed

Line 71-76: Inserted text Two systematic reviews by Fryer et al. (2016) and Wray et al.(2018) have investigated the effects of self-management interventions, and found that self-management interventions may improve quality of life and hence self-efficacy post-stroke [24,25]. However, they investigated the effect of self-management interventions on the entire adult population with stroke, leaving a lack of knowledge on which core elements of self-management interventions are effective for people with stroke over the age of 65.

Line 78-79: Neurological literature indicates overlapping terminology and concepts, for instance, between coping, adaption and self-management Sentence has been removed

Line 81-82: focusing on impairment level according to the International Classification of Function (ICF) Text has been removed

Line 84-85: by an approach targeting participation level by Text has been removed

Line 86-96: Two systematic reviews by Fryer et al. and Wray et al. have investigated the effects of self-management interventions on the entire adult population with stroke [24,25], and found that self-management interventions may improve quality of life and hence self-efficacy post-stroke. However, the results are associated with considerable uncertainty, including uncertainty regarding which core elements of self-management interventions are effective [24,25]. Despite the limitations of existing evidence, Parke et al. recommend providing people with stroke (and their caregivers) rehabilitation with emotional and social self-management support in order to improve outcomes such as activities of daily living and reintegration into the community [26]. Old age pensioners are an especially vulnerable group, prone to loneliness, social isolation and lack of social reintegration due to bodily frailty, inactivity and difficulties in terms of social relationships after leaving the labour market. Text has been removed

Comment 3) The Discussion section should also be shortened, especially the subsections on "limitations" and "implications for future research".

The Discussion has been shortened. We have omitted general descriptive details. Our changes:

Line 272-274: However, the two concepts differ, as self-care focuses on the performance of daily activities independently of the healthcare system, while self-management involves knowledge, self-regulation and social facilitation to manage a life with stroke or to engage in healthy behaviours Sentence has been removed

Line 275-279: In the development of complex interventions, the use of theory is emphasised [76] as evidence suggests that theoretically informed interventions lead to better outcomes [77]. Using behaviour change theory to develop interventions also provides a way of understanding an intervention’s effectiveness, or lack thereof [78]. The characterization of theoretical foundation may not be exhaustive. Three sentences have been removed

Line 287-299: According to the Medical Research Council Framework (MRC Framework), a complex intervention is an intervention that includes several interacting components. These components interact with the context in which the intervention is delivered and involve behaviours of both those delivering and receiving an intervention aimed at changing one or more outcomes [76,79]. The eleven included studies show complexity in the interacting processes, behavioural patterns and target outcomes. It is therefore reasonable to characterise all of the included studies as complex interventions. According to the MRC Framework, the development of a complex intervention should be based on a four-stage process of ‘develop-test-evaluate-implement’ [76]. Performing a properly randomized clinical trial with a complex intervention thus requires preparation to uncover salient key features and ensure the targeted measurement methods. Only one study used the MRC Framework to develop and feasibility test their intervention before implementing info their randomized controlled trial [69]. Such a process could have helped most of the other included studies achieve a clearer, more focused research design and improved the methodological quality. Text has been removed

Line 305-306: Changing self-management behaviours is useless if the behaviours do not improve outcomes such as quality of life Sentence has been removed

Line 307-308: Inserted text Researchers should be aware of this divergence in process and outcome-oriented thinking regarding self-management in future research.

Line 341-343: This divergence in process and outcome-oriented thinking highlights uncertainties regarding the definition of self-management in stroke rehabilitation. Sentence has been removed

Line 344-350: Inserted text All the interventions in the included studies could be regarded complex [76]. According to the Medical Research Council (MRC) Framework, the development of a complex intervention should be based on a four-stage process of ‘develop-test-evaluate-implement’ [76]. Only one study used the MRC Framework to develop and feasibility test their intervention before implementing into their randomized controlled trial [69]. Future research in complex interventions should be designed and feasibility tested with implementation in mind at the very start as it is likely to improve the chances of embedding the new intervention into routine practice.

Line 352-354: Post-stroke depression occurs in one third of stroke survivors [84]. It is a matter of concern that only short-termed effects, if any, could be revealed. However, since the quality of the included studies was low, no firm conclusions can be drawn. Two sentences have been removed

Line 355-356: Inserted text especially in the elderly population

Line 360-361: Thus, self-management interventions may be unable to promote physical activity sufficiently for those patients who would need it the most. Sentence has been removed

Line 365-366: Inserted text In future research an increased focus on involving relatives and network in stroke self-management interventions is recommended.

Line 367-377: However, activities of daily living and physical activity are prerequisites for participation. It is worth noting that activity and participation vary between different cultures. Therefore, we may reasonably assume that participation is to some extent reflected in these instruments. Moreover, consensus on participation measures for patients with stroke is lacking [90]. This review differed from the systematic reviews by Fryer et al. (2016) and Wray et al. (2018) in terms of their inclusion criteria, namely the age group and the outcome measures [24,25]. Although efforts have been made to make meta-analyses in the reviews mentioned – unlike the present study – the diversity of the self-management interventions and outcome measures makes pooling data across studies and synthesising the available evidence questionable. However, there is agreement that more studies are needed to reveal the core elements of effective self-management interventions for adults with stroke. Text has been removed

Best regards,

Sedsel Kristine Stage Pedersen

First author

Reviewer 2 Report

In this manuscript, the authors undertake a systematic review to evaluate the effectiveness of self-management interventions for older people with stroke (mean age 65 or older in both intervention and control group) on seven psychosocial outcomes: self-management, self-efficacy, quality of life, depression activities of daily living, active lifestyle, and other. An initial search of PubMed, Embase and PsychInfo for randomized controlled trials and an assortment of relevant keywords initially yielded 820 articles, which was narrowed down to 11 articles that met rigorous eligibility criteria to be included in the systematic review. Given the heterogeneity of interventions, time of follow-up, and outcome measures, a meta-analysis was deemed infeasible. A narrative examination of these articles suggested that self-management interventions for people with stroke over age 65 may be beneficial for the outcome measures of self-management self-efficacy, quality of life, activity of daily living, and other psychosocial outcomes. The poor quality and heterogeneity of the studies led the authors’ to be very cautious in drawing robust conclusions.

This systematic review addresses an important issue regarding the effect of self-management interventions for elderly people with stroke. The authors were rigorous in vetting articles for eligibility for inclusion in the systematic review and were appropriately cautious in drawing conclusions. Below are some suggestions for further strengthening the manuscript.

  1. Table 2 needs to be improved in format and clarity. It would help if the table could be in landscape orientation with the text in each column left justified. Consider ways to make the text in each cell more concise. Consider separating multiple headings in one column into two or more columns for clarity. For example, have separate columns for sample size, mean age and SD, study design, population, aim, follow-up time points, risk of bias, etc. Also consider a more concise way of presenting the outcome measures, such as a separate table (or note at bottom of table) of all outcome measures and their acronyms, so that only acronyms referring to measures can be included in the table. The number of decimals used for mean age (and SD) should be consistent – at present the number of decimals ranges from zero to two. Mean age and SD should be consistently aligned within a column that is wide enough so that they do not wrap around. Such edits should help make the table much easier to read.
  2. Please check Table 2 to ensure that the country is listed for each article. It is missing in the first column for the 2020 article by Fu et al. but appears in a different column listing measures. It would help to be consistent in providing this information in the same column.
  3. Please check Table 2 to ensure that the mean age of intervention and control groups are consistently listed for each article. The mean age of the control group in the 2015 article by Bishop et al. is missing.
  4. Table 1. “PudMed” should be changed to “PubMed”
  5. Lines 107, 133-134. Please clarify what the “estimated coefficient” refers to. The authors mention improvement, which suggests a pre-post comparison. Is it a regression coefficient or a mean pre-post difference?
  6. Line 159. Something is missing from the Confidence Interval (CI). It should be a range.
  7. Line 160. “difference” should be changed to “differences”
  8. Lines 171, 175. It would help to be consistent in presenting Confidence Intervals, instead of presenting a range (-1.4 to 0.69) in one case and two values separated by a comma (-0.2,0.8) in another case.
  9. Discussion. It would help if the authors would comment on whether any study designs other than randomized controlled trials could have been included in the systematic review, given that they ended up conducting a narrative synthesis rather than a meta-analysis.

Author Response

Dear reviewer 2

Thanks for your comments to strengthen our manuscript. We have made the corrections below according to your comments.

Comment 1: Table 2 needs to be improved in format and clarity. It would help if the table could be in landscape orientation with the text in each column left justified. It would help if the table could be in landscape orientation with the text in each column left justified.. Consider separating multiple headings in one column into two or more columns for clarity. For example, have separate columns for sample size, mean age and SD, study design, population, aim, follow-up time points, risk of bias, etc. Also consider a more concise way of presenting the outcome measures, such as a separate table (or note at bottom of table) of all outcome measures and their acronyms, so that only acronyms referring to measures can be included in the table. The number of decimals used for mean age (and SD) should be consistent – at present the number of decimals ranges from zero to two. Mean age and SD should be consistently aligned within a column that is wide enough so that they do not wrap around. Such edits should help make the table much easier to read.

We agree that table 2 needs to be improved. We have redesigned the table to make it more readable. Our changes:

  • Column 1 (Authors, Title, Year, Journal, Country): The column has been edited to be consistent throughout
  • Column 2 (Sample size, Age Mean (SD): The column has been divided into two
  • Column 3 (Study design, Population, Aim: The column has been divided into three
  • Column 5 (Intervention/comparator): The column has been edited to be consistent throughout
  • Column 6 (Psychosocial Outcome measures): The column has been edited, and presented according to the overall division of outcome measures i) self-management, ii) self-efficacy, iii) quality of life, iv) depression, v) activity of daily living, vi) active lifestyle and vii) other measures. Furthermore, the column has been supplemented with an overview of the measurements in the online supplementary information (The table is referred to in line 173-174, and is attached to the supplementary materiel)
  • Column 7 (Follow-up time point/Risk of bias): Follow up time has been elaborated, and risk of bias has been removed as it is presented elsewhere in the manuscript

Comment 2: Please check Table 2 to ensure that the country is listed for each article. It is missing in the first column for the 2020 article by Fu et al. but appears in a different column listing measures. It would help to be consistent in providing this information in the same column.

The columns have been edited to be consistent throughout. The missing years in the articles by Fu et al. and Hjelle et al. have been added.

Comment 3: Please check Table 2 to ensure that the mean age of intervention and control groups are consistently listed for each article. The mean age of the control group in the 2015 article by Bishop et al. is missing.

The mean age has been reviewed and edited to be consistent throughout. However, the articles by Bishop et al. and Lo et al. only reported a total mean age for both the intervention and control group,

Comment 4: Table 1. “PudMed” should be changed to “PubMed”

The spelling has been corrected.

Comment 5: Lines 107, 133-134. Please clarify what the “estimated coefficient” refers to. The authors mention improvement, which suggests a pre-post comparison. Is it a regression coefficient or a mean pre-post difference?

Line 97 + 121 The authors behind the results have used a generalised estimating equation (GEE) model to assess differential change of the outcomes across the time points between the two groups. We interpret the result to be "estimated mean population difference" and have corrected the “estimated coefficient” to “mean difference”.

Comment 6: Line 159. Something is missing from the Confidence Interval (CI). It should be a range.

Line 144: The Confidence Interval (CI) has been corrected.

Comment 7: Line 160. “difference” should be changed to “differences”

Line 146: The spelling has been corrected.

Comment 8: Lines 171, 175. It would help to be consistent in presenting Confidence Intervals, instead of presenting a range (-1.4 to 0.69) in one case and two values separated by a comma (-0.2,0.8) in another case.

Line 156: The Confidence Intervals have been reviewed and edited to be consistent throughout. “To” have been replaced with “,”.

Comment 9: Discussion. It would help if the authors would comment on whether any study designs other than randomized controlled trials could have been included in the systematic review, given that they ended up conducting a narrative synthesis rather than a meta-analysis.

Line 268-271: The following text has been inserted Since it ended with a narrative synthesis rather than a meta-analysis in this review, other study designs could have been usefully included to identify additional dimensions and strengthen the evidence of 'what works' in self-management interventions to elderly people over the age of 65.

Best regards,

Sedsel Kristine Stage Pedersen

First author

Reviewer 3 Report

The paper is on a very interesting topic. 

It is well designed and well written. 

The systematic research is complete.

Author Response

Dear reviewer 3

Thanks for your comments. We are pleased to hear that you liked the topic and the manuscript.

Best regards,

Sedsel Kristine Stage Pedersen

First author